# Lorentz-Equivariant Geometric Algebra Transformers for High-Energy Physics

**Jonas Spinner**[*]
Heidelberg University
j.spinner@thphys.uni-heidelberg.de

**Victor Bresó**[*]
Heidelberg University
v.breso@thphys.uni-heidelberg.de

**Pim de Haan**
Qualcomm AI Research[†]

**Tilman Plehn**
Heidelberg University

**Jesse Thaler**
MIT / IAIFI

**Johann Brehmer**
Qualcomm AI Research[†]

## Abstract

Extracting scientific understanding from particle-physics experiments requires solving diverse learning problems with high precision and good data efficiency. We propose the Lorentz Geometric Algebra Transformer (L-GATr), a new multi-purpose architecture for high-energy physics. L-GATr represents high-energy data in a geometric algebra over four-dimensional space-time and is equivariant under Lorentz transformations, the symmetry group of relativistic kinematics. At the same time, the architecture is a Transformer, which makes it versatile and scalable to large systems. L-GATr is first demonstrated on regression and classification tasks from particle physics. We then construct the first Lorentz-equivariant generative model: a continuous normalizing flow based on an L-GATr network, trained with Riemannian flow matching. Across our experiments, L-GATr is on par with or outperforms strong domain-specific baselines.

## 1 Introduction

In the quest to understand nature on the most fundamental level, machine learning is omnipresent [23]. Take the most complex machine ever built: at CERN's Large Hadron Collider (LHC), protons are accelerated to close to the speed of light and interact; their remnants are recorded by various detector components, totalling around $10^{15}$ bytes of data per second [24]. These data are filtered, processed, and compared to theory predictions, as we sketch in Fig. 1. Each step of this pipeline requires making decisions about high-dimensional data. More often than not, these decisions are rooted in machine learning, increasingly often deep neural networks [12, 13, 20, 32, 44, 49, 52, 66]. This approach powers most measurements in high-energy physics, culminating in the Higgs boson discovery in 2012 [5, 30].

High-energy physics analyses put stringent requirements on network architectures. They need to be able to represent particle data and have to be expressive enough to learn complex relations in high-dimensional spaces precisely. Moreover, training data often come from precise theory computations and complex detector simulations, both of which require a considerable computational cost; architectures therefore need to be data efficient. Generative models of particle-physics data face additional challenges: because of detector boundaries and selection cuts, densities frequently feature sharp edges; at the same time, it is often important to model low-density tails of distributions precisely over multiple orders of magnitude of probability densities.

Off-the-shelf architectures originally developed for vision or language are popular starting points for

---

[*]Equal contribution
[†]Qualcomm AI Research is an initiative of Qualcomm Technologies, Inc.

38th Conference on Neural Information Processing Systems (NeurIPS 2024).

high-energy physics applications [19, 37], but do not satisfy these goals reliably. We argue that this is because they do not make systematic use of the rich structure of the data. Particle interactions are governed by quantum field theories and respect their symmetries, notably the Lorentz symmetry of special relativity [39, 63]. First Lorentz-equivariant architectures have recently been proposed [10, 42, 68], but they are limited to specific applications and not designed with a focus on scalability.

In this work, we introduce the Lorentz Geometric Algebra Transformer (L-GATr), a new general-purpose network architecture for high-energy physics. It is based on three design choices. First, L-GATr is equivariant with respect to the Lorentz symmetry.[3] It supports partial and approximate symmetries as found in some high-energy physics applications through symmetry-breaking inputs. Second, as representations, L-GATr uses the geometric (or Clifford) algebra over the four-vectors of special relativity. This algebra is based on the scalar and four-vector properties that LHC data are naturally parameterized in and extends them to higher orders, increasing the network capacity. Finally, L-GATr is a Transformer. It supports variable-length inputs, as found in many LHC problems, and even large models can be trained efficiently. Because it computes pairwise interactions through scaled dot-product attention, for which there are highly optimized backends like Flash Attention [33], the architecture scales particularly well to problems with many tokens or particles.

L-GATr is based on the Geometric Algebra Transformer architecture [14, 36], which was designed for non-relativistic problems governed by the Euclidean symmetry E(3) of translations, rotations, and reflections. Our L-GATr architecture generalizes that to relativistic scenarios and the Lorentz symmetry. To this end, we develop several new network layers, including a maximally expressive Lorentz-equivariant linear map, a Lorentz-equivariant attention mechanism, and Lorentz-equivariant layer normalization.

In addition to the general-purpose architecture, we develop the first (to the best of our knowledge) Lorentz-equivariant generative model. We construct a continuous normalizing flow with an L-GATr denoising network and propose training it with a Riemannian flow matching approach [25]. This not only lets us train the model in a scalable way, but also allows us to encode more aspects of the problem geometry into the model: we can even hard-code phase-space boundaries, which are commonplace in high-energy physics.

We demonstrate L-GATr on three particle-physics applications. We first train neural surrogates for quantum field theoretic amplitudes, a regression problem with high demands on precision. Next, we train classification models and evaluate L-GATr on the popular benchmark problem of top tagging. Finally, we turn to the generative modelling of reconstructed particles, which can make the entire analysis pipeline substantially more efficient. The three applications differ in the role they play in the LHC analysis pipeline (see Fig. 1), data, and learning objective, highlighting the versatility of L-GATr. We find that L-GATr is on par with or outperforms strong domain-specific baselines across all problems, both in terms of performance and data efficiency.

Our implementation of L-GATr is available at `https://github.com/heidelberg-hepml/lorentz-gatr`.

## 2 Background and related work

**High-energy physics** In Fig. 1 we sketch the typical data-analysis pipeline in particle physics. Its central idea is to take the data collected in the detectors as well as the predictions from different theories of physics, process both in parallel, and ultimately compare their predictions. The pipeline includes various steps, including the computation of scattering probabilities or amplitudes in mathematical frameworks called quantum field theories, the Monte-Carlo sampling from the theory, the simulated interaction of particles with the detector, the dimensionality reduction of the raw detector output to a small number of observables, the data filtering to extract only collisions of interest, and the statistical analysis of whether two predictions are consistent.

What most steps in this pipeline have in common is the notion of *particles*, the main representation of data in high-energy physics. A particle is characterized by a discrete type label, an energy $E \in \mathbb{R}$, and a spatial momentum $\vec{p} \in \mathbb{R}^3$. Types include fundamental particles like electrons, photons, quarks, and

---

[3]One could extend L-GATr to the full Poincaré symmetry, which additionally includes space-time translations. However, this is not necessary for most particle-physics applications, as usually only the momentum, and not the absolute position, of particles is of interest. A key exception is the study of long-lived particles.

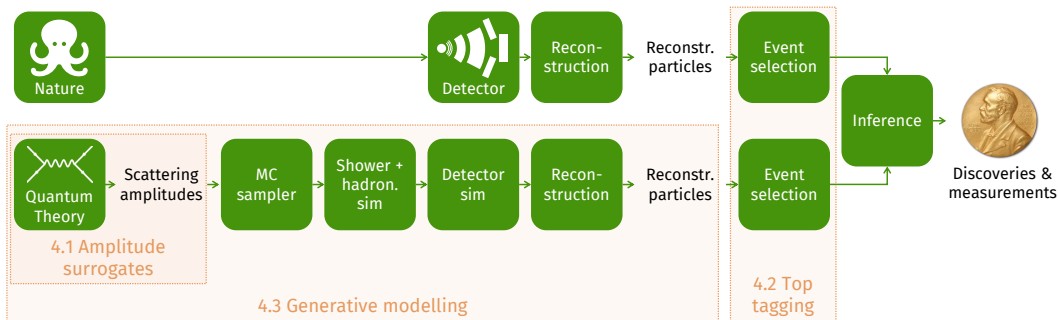

Figure 1: Schematic view of the data-analysis workflow in high-energy physics. Measurements (top) are processed in parallel with simulated data (bottom); their comparison is ultimately the basis for most scientific conclusions. In orange, we show how the three applications of L-GATr we experiment with in this paper fit into this workflow. The architecture is also applicable in several other stages, including reconstruction and inference.

gluons, composite particles like protons, as well as reconstructed objects like "jets" [70] or "particle-flow candidates" [71], which are the outputs of complex reconstruction algorithms. The energy and spatial momentum of a particle are conveniently combined into a *four-momentum* $p = (E, \vec{p})$.

The laws of fundamental physics [41, 75] are invariant with respect to the choice of an inertial reference frame [39, 63]: they do not change under rotations and boosts from one un-accelerated reference frame into another.[4] Together, these transformations form the *special orthochronous Lorentz group* $\mathrm{SO}^+(1,3)$.[5] This group is the connected component of the orthogonal group on the four-vector space $\mathbb{R}^{1,3}$ with Minkowski metric $\mathrm{diag}(+1, -1, -1, -1)$ [61]. Lorentz transformations mix temporal and spatial components. Space and time should therefore not be considered as separate concepts, but rather as components of a four-dimensional space-time. Particle four-momenta are another instance of this: they transform in the vector representation of the Lorentz group as $p^\mu \to p'^\mu = \sum_\nu \Lambda^\mu_\nu p^\nu$ for $\Lambda \in \mathrm{SO}^+(1,3)$, with the Lorentz transformation mixing energy and spatial momentum.

**Geometric deep learning** The central tenet of geometric deep learning [15, 31] is to embed the known structure of a problem into the architecture used to solve it, instead of having to learn it completely from data. The key idea is that of *equivariance* to symmetry groups: when the inputs $x$ to a network $f$ are transformed with a symmetry transformation $g$, the outputs should transform under the same element of the symmetry group, $f(g \cdot x) = g \cdot f(x)$, where $\cdot$ denotes the group action. What is known as "equivariance" in machine learning is often called "covariance" in physics [28].

**GATr** Our work is rooted in the Geometric Algebra Transformer (GATr) [14, 36], a network architecture that is equivariant to $\mathrm{E}(3)$, the group of non-relativistic translations, rotations, and reflections. GATr represents inputs, hidden states, and outputs in the geometric (or Clifford) algebra $\mathbb{G}_{3,0,1}$ [29, 43, 67]. A geometric algebra extends a base space like $\mathbb{R}^3$ to higher orders and adds a bilinear map known as the *geometric product*. We provide a formal introduction in Appendix A. What matters in practice is that this vector space can represent various 3D geometric objects. Brehmer et al. [14] develop different layers for this representation and combine them in a Transformer architecture [73]. For L-GATr, we build on the GATr blueprint, but re-design all components such that they can represent four-momenta and are equivariant with respect to Lorentz transformations.

**Lorentz-equivariant architectures** Recently, some Lorentz-equivariant architectures have been proposed. Most closely related to this work is the Clifford Group Equivariant Neural Networks (CGENN) by Ruhe et al. [68] and their extensions to simplical complexes [54] and steerable convolutions [78]. Like us, they use the geometric algebra over four-vectors. While they also use Lorentz-equivariant linear maps and geometric products, our architectures differ in a number of ways. In particular, they propose a message-passing graph neural network, while we build a Transformer archi-

---

[4]Allowing for accelerating reference frames would bring us to the general theory of relativity, which is irrelevant for particle physics experiment as long as they are not performed close to a black hole.

[5]"Special" and "orthochronous" here mean that spatial and temporal reflections are not considered as symmetries. In fact, the fundamental laws of nature are *not* invariant under those transformations, an effect known as $P$-violation and $T$-violation.

tecture based on dot-product attention.

Other Lorentz-equivariant architectures include LorentzNet [42] and the Permutation Equivariant and Lorentz Invariant or Covariant Aggregator Network (PELICAN) [10]. Both are message-passing graph neural network as well. Given a set of four-vectors, PELICAN computes all pairwise inner products, which are Lorentz invariants, and then processes them with a permutation-equivariant architecture. LorentzNet maintains scalar and four-vector representations and updates them with a graph attention mechanism similar to the one proposed by Villar et al. [74].

**Flow matching**  Continuous normalizing flows [26] are a class of generative models that push a sample from a base density through a transformation defined by an ordinary differential equation.

Specifically, the evolution of a point $x \in \mathbb{R}^d$ is modelled as a time-dependent flow $\psi_t : \mathbb{R}^d \to \mathbb{R}^d$ with $\frac{\mathrm{d}}{\mathrm{d}t}\psi_t(x) = u_t(\psi_t(x)), \psi_0(x) = x$, where $u_t$ is a time-dependent vector field.

Conditional flow matching [3, 53] is a simple and scalable training algorithm for continuous normalizing flows that does not require the simulation of trajectories during training. Instead, the objective is to match a vector field $v_t(x)$, parametrized by a neural network, onto a conditional target vector field $u_t(x|x_1)$ along a conditional probability path $p_t(x|x_1)$, minimizing the loss $\mathcal{L}_{\mathrm{CFM}} = \mathbb{E}_{t \sim \mathcal{U}[0,1], x_1 \sim q(x_1), x \sim p_t(x|x_1)} \|v_t(x) - u_t(x|x_1)\|^2$, where $x_1 \sim q(x_1)$ are samples from the base distribution.

Choosing a well-suited probability path and corresponding target vector field can substantially improve the data efficiency and sampling quality. A principled approach to this choice is Riemannian flow matching (RFM) [25]. Instead of connecting target and latent space points by straight lines in Euclidean space, RFM proposes to choose probability paths based on the metric of the manifold structure of the data space. If available in closed form, they propose to use geodesics as probability paths, which corresponds to optimal transport between base and data density.

## 3   The Lorentz Geometric Algebra Transformer (L-GATr)

### 3.1   Lorentz-equivariant architecture

**Geometric algebra representations**  The inputs, hidden states, and outputs of L-GATr are variable-size sets of tokens. Each token consists of $n$ copies of the geometric algebra $\mathbb{G}_{1,3}$ and $m$ additional scalar channels.

The geometric algebra $\mathbb{G}_{1,3}$ is defined formally in Appendix A. In practice, $\mathbb{G}_{1,3}$ is a 16-dimensional vector space that consists of multiple subspaces (or grades). The 0-th grade consists of scalars that do not transform under Lorentz transformations, for instance embeddings of particle types or regression amplitudes. The first grade contains space-time four-vectors such as the four-momenta $p = (E, \vec{p})$. The remaining grades extend these objects to higher orders (i. e. antisymmetric tensors), increasing expressivity. In addition, the geometric algebra defines a bilinear map, the geometric product $\mathbb{G}_{1,3} \times \mathbb{G}_{1,3} \to \mathbb{G}_{1,3}$, which contains both the space-time inner product and a generalization of the Euclidean cross product.

This representation naturally fits most LHC problems, which are canonically represented as sets of particles, each parameterized with type information and four-momenta. We represent each particle as a token, store the particle type as a one-hot embedding in the scalar channels and the four-momentum in the first grade of the geometric algebra.

**Lorentz-equivariant linear layers**  We define several new layers that have both $\mathbb{G}_{1,3}$ and additional scalar representations as inputs and outputs. For readability, we will suppress the scalar channels in the following. We require each layer $f(x)$ to be equivariant with respect to Lorentz transformations $\Lambda \in \mathrm{SO}^+(1,3)$: $f(\Lambda \cdot x) = \Lambda \cdot f(x)$, where $\cdot$ denotes the action of the Lorentz group on the geometric algebra (see Appendix A). Lorentz equivariance strongly constrains linear maps between geometric algebra representations:[6]

---

[6]$\mathrm{O}(1,3)$-equivariant linear maps are restricted to the first sum; the additional equivariance under reflections forbids the multiplication with the pseudoscalar.

**Proposition 1.** *Any linear map* Linear $: \mathbb{G}_{1,3} \to \mathbb{G}_{1,3}$ *that is equivariant to* $\mathrm{SO}^{+}(1,3)$ *is of the form*

$$\mathrm{Linear}(x) = \sum_{k=0}^{4} v_k \langle x \rangle_k + \sum_{k=0}^{4} w_k e_{0123} \langle x \rangle_k \tag{1}$$

*for parameters* $v, w \in \mathbb{R}^5$. *Here* $e_{0123}$ *is the pseudoscalar, the unique highest-grade basis element in* $\mathbb{G}_{1,3}$; $\langle x \rangle_k$ *is the blade projection of a multivector, which sets all non-grade-$k$ elements to zero.*

We show this in Appendix A. In our architecture, linear layers map between multiple input and output channels. There are then ten learnable weights $v_k, w_k$ for each pair of input and output $\mathbb{G}_{1,3}$ channels (plus the usual weights for linear maps between the additional scalar channels).

**Lorentz-equivariant non-linear layers**   We define four additional layers, all of which are manifestly Lorentz-equivariant. The first is the scaled dot-product attention

$$\mathrm{Attention}(q, k, v)_{i'c'} = \sum_i \mathrm{Softmax}_i \left( \sum_{c=1}^{n_c} \frac{\langle q_{i'c}, k_{ic} \rangle}{\sqrt{16 n_c}} \right) v_{ic'} \,, \tag{2}$$

where the indices $i, i'$ label tokens, $c, c'$ label channels, $n_c$ is the number of channels, and $\langle \cdot, \cdot \rangle$ is the $\mathbb{G}_{1,3}$ inner product. This inner product can be rewritten as a pre-computed list of signs and a Euclidean inner product, which is why we can compute the attention mechanism with efficient backends developed for the original Transformer architecture, for instance Flash Attention [33]. This is key to the good scalability of L-GATr, which we will demonstrate later.

When defining a normalization layer, we have to be careful: in the $\mathbb{G}_{1,3}$ inner product, cancellations between positive-norm directions and negative-norm directions can lead to norm values much smaller than the scale of the individual components; dividing by the norm then risks blowing up the data. These cancellations are an unavoidable consequence of the geometry of space-time. We mitigate this issue by using the grade-wise absolute value of the inner product in the norm

$$\mathrm{LayerNorm}(x) = x \Big/ \sqrt{\frac{1}{n_c} \sum_{c=1}^{n_c} \sum_{k=0}^{4} \left| \left\langle \langle x_c \rangle_k, \langle x_c \rangle_k \right\rangle \right| + \epsilon} \,, \tag{3}$$

applying an absolute value around each grade of each multivector channel $\langle x_c \rangle_k$. Here $\epsilon > 0$ is a constant that further numerically stabilizes the operation. This normalization was proposed by De Haan et al. [36] for $\mathrm{E}(3)$-invariant architectures, we adapt it to the Lorentz-equivariant setting.

We also use the geometric product $\mathrm{GP}(x, y) = xy$ defined by the geometric algebra $\mathbb{G}_{1,3}$. Finally, we use the scalar-gated GELU [46] nonlinearities $\mathrm{GatedGELU}(x) = \mathrm{GELU}(\langle x \rangle_0) x$, as proposed by Brehmer et al. [14].

**Transformer architecture**   We combine these layers into a Transformer architecture [73, 76]:

$$\bar{x} = \mathrm{LayerNorm}(x) \,,$$
$$\mathrm{AttentionBlock}(x) = \mathrm{Linear} \circ \mathrm{Attention}(\mathrm{Linear}(\bar{x}), \mathrm{Linear}(\bar{x}), \mathrm{Linear}(\bar{x})) + x \,,$$
$$\mathrm{MLPBlock}(x) = \mathrm{Linear} \circ \mathrm{GatedGELU} \circ \mathrm{Linear} \circ \mathrm{GP}(\mathrm{Linear}(\bar{x}), \mathrm{Linear}(\bar{x})) + x \,,$$
$$\mathrm{Block}(x) = \mathrm{MLPBlock} \circ \mathrm{AttentionBlock}(x) \,,$$
$$\mathrm{L\text{-}GATr}(x) = \mathrm{Linear} \circ \mathrm{Block} \circ \mathrm{Block} \circ \cdots \circ \mathrm{Block} \circ \mathrm{Linear}(x) \,.$$

This L-GATr architecture is structurally similar to the original GATr architecture [14], but the representations, linear layers, attention mechanism, geometric product, and normalization layer are different to accommodate the different nature of the data and different symmetry group.

**Lorentz symmetry breaking**   While fundamental physics is (to the best of our knowledge) symmetric under Lorentz transformations, the LHC measurement process is not. The direction of the proton beams presents the most obvious violation of this symmetry. Smaller violations are due to the detector resolution: particles hitting the central part of the detector (orthogonal to the beam in the detector rest frame) are typically reconstructed with a higher precision than those emerging at a narrow angle

to the beam. Even smaller violations come, for instance, from individual defunct detector elements. Solving some tasks may therefore benefit from a network that can break Lorentz equivariance.

L-GATr supports such broken or approximate symmetries by including the symmetry-breaking effects as additional inputs into the network. Concretely, whenever we analyze reconstruction-level data, we include the beam directions; see Appendix C. This approach combines the strong inductive biases of a Lorentz-equivariant architecture with the ability to learn to break the symmetry when required.

## 3.2 Lorentz-equivariant flow matching

In addition to regression and classification models, we construct a generative model for particle data. Besides the strict requirements on precision, flexibility, and data efficiency, generative models of LHC data need to be able to address sharp edges and long tails in high-dimensional distributions.

We develop a continuous normalizing flow based on an L-GATr vector field and train it with Riemannian flow matching (RFM) [25]. This approach has several compelling properties: training is simulation-free and scalable and the generative model is Lorentz-equivariant.[7] In addition, the RFM approach allows us to deal with sharp edges and long tails in a geometric way: we parameterize the reachable four-momentum space for each particle as a manifold and use geodesics on this manifold as probability paths from base samples to data points.

**Probability paths perfect for particles**  Concretely, reconstructed particles $p = (E, \vec{p})$ are often required to satisfy constraints of the form $p_1^2 + p_2^2 \geq p_{T\,\mathrm{min}}^2$ and $p^2 > 0$. Following Refs. [17, 18, 45, 48], we parameterize this manifold with physically motivated coordinates $y = (y_m, y_p, \eta, \phi)$. These variables form an alternative basis for the particle four-momenta and are defined through the map

$$p = (E, p_x, p_y, p_z) = f(y) = \left( \sqrt{m^2 + p_T^2 \cosh^2 \eta}, \; p_T \cos\phi, \; p_T \sin\phi, \; p_T \sinh\eta \right), \quad (4)$$

where $m^2 = \exp(y_m)$ and $p_T = p_{T,\mathrm{min}} + \exp(y_p)$. This basis is better aligned with the physically relevant properties of particles in the context of a collider experiment: $\eta$ and $\phi$ represent the angle in which a particle is moving, $y_p$ is a measure of the momentum with which it moves away from the collision, and $y_m$ is related to its mass.

We define a constant diagonal metric in the coordinates $y$ and use the corresponding geodesics as probability paths. This Riemannian manifold is geodesically convex, meaning any two points are connected by a unique geodesic, and geodesically complete, meaning that paths thus never enter four-momentum regions forbidden by the phase-space cuts. By also running the ordinary differential equation (ODE) solver in these coordinates, we guarantee that each sample satisfies the four-momentum constraints. As an added benefit, this choice of metric compresses the high-energy tails of typical particle distributions and thus simplifies learning them correctly.

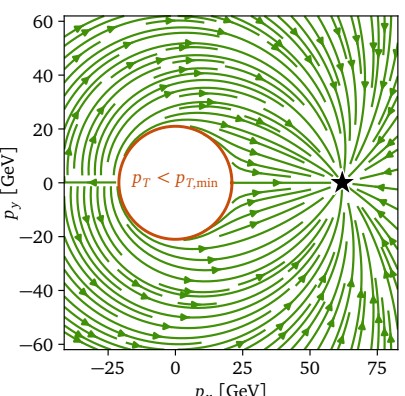

Figure 2: Target vector field for Riemannian flow matching. Our choice of metric space guarantees that the generative model respects phase-space boundaries (red circle).

In Fig. 2, we show target probability paths generated in this way. Our approach ensures that none of the trajectories pass through the phase-space region $p_T < p_{T,\mathrm{min}}$, where the target density does not have support; instead, the geodesics lead around this problematic region.

---

[7]Strictly speaking, only the map from the base density to data space is equivariant with respect to the full Lorentz group. The base density and thus also the density of the generative model are only invariant with respect to rotations. This is because the group of boosts is not compact: it is impossible to define a properly normalized density that assigns the same probability to every boosted data variation. In theory, one could define a fully Lorentz-invariant base measure; then the flow would define a Lorentz-invariant measure that would not be normalizable—good luck with that. In practice, compact subsets of the orbits, for instance characterized by a limited range of the center-of-mass momentum, suffice. All of this is in analogy to "E(3)-invariant" generative models [47], which are strictly only invariant to rotations, but not to (non-compact) translations.

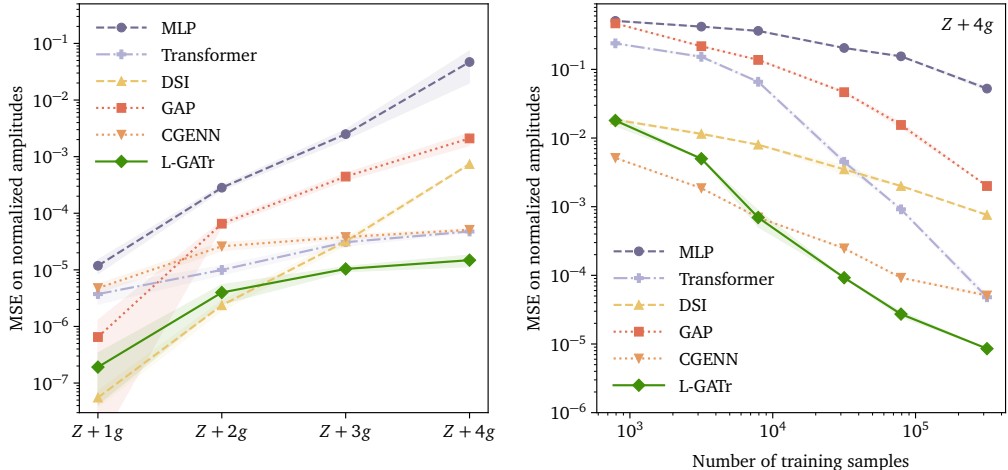

Figure 3: Amplitude surrogates. **Left**: Surrogate error for processes of increasing particle multiplicity and complexity, training on the full dataset of $4 \cdot 10^5$ samples. L-GATr outperforms the baselines, especially at more complex processes. **Right**: Surrogate error as a function of the training dataset size.

## 4   Experiments

We now demonstrate L-GATr in three applications. Each addresses a different problem in the data-analysis pipeline sketched in Fig. 1.

### 4.1   Surrogates for QFT amplitudes

**Problem**   We first demonstrate L-GATr as a neural surrogate for quantum field theoretical ampli-tudes [6–8, 59, 60], the core of the theory predictions that LHC measurements are compared to. These amplitudes describe the (un-normalized) probability of interactions of fundamental particles as a function of their four-momenta. As this is a fundamental interaction and does not include the measurement process, it is exactly Lorentz-invariant. Evaluating them is expensive, on the one hand because it requires solving complex integrals, on the other hand because the number of relevant terms combinatorially grows with the number of particles. Neural surrogates can greatly speed up this process and thus enable better theory predictions, but accurately modelling the amplitudes of high-multiplicity processes has been challenging.

As example processes, we study $q\bar{q} \to Z + ng$, the production of a $Z$ boson with $n = 1, \ldots, 4$ additional gluons from a quark-antiquark pair. For each gluon multiplicity, we train a L-GATr model to predict the amplitude as a function of the four-momenta of the initial and final particles.[8] The generation of the training data and the precise setup of the learning problem are described in Appendix C. We compare L-GATr to various baselines, including the Lorentz-equivariant message-passing architecture CGENN [68], a Transformer [73], and DSI, a baseline based on the Deep Sets framework [77] that we develop ourselves; we describe it in detail in Appendix B.

**Surrogate quality**   L-GATr consistently approximates the amplitudes with high precision, as we show in the left panel of Fig. 3. For a small number of particles, it is slightly worse than our own baseline DSI, but it scales much better to a large number of particles, where it outperforms all other methods. This is exactly the region in which neural surrogates could have the highest impact.

**Data efficiency**   In the right panel of Fig. 3 we study the data efficiency of the different architectures. We find that L-GATr is competitive at any training data size, combining the small-data advantages of its strong inductive biases and the big-data advantages of its Transformer architecture.

---

[8]We also experimented with training a single L-GATr model to learn the amplitudes of all processes jointly, finding a similar performance.

| Model | Accuracy | AUC | $1/\epsilon_B$ ($\epsilon_S = 0.5$) | $1/\epsilon_B$ ($\epsilon_S = 0.3$) |
|---|---|---|---|---|
| TopoDNN [49] | 0.916 | 0.972 | – | 295 ± 5 |
| LoLa [16] | 0.929 | 0.980 | – | 722 ± 17 |
| P-CNN [1] | 0.930 | 0.9803 | 201 ± 4 | 759 ± 24 |
| $N$-subjettiness [62] | 0.929 | 0.981 | – | 867 ± 15 |
| PFN [51] | 0.932 | 0.9819 | 247 ± 3 | 888 ± 17 |
| TreeNiN [58] | 0.933 | 0.982 | – | 1025 ± 11 |
| ParticleNet [64] | 0.940 | 0.9858 | 397 ± 7 | 1615 ± 93 |
| ParT [65] | 0.940 | 0.9858 | 413 ± 16 | 1602 ± 81 |
| LorentzNet* [42] | 0.942 | 0.9868 | 498 ± 18 | 2195 ± 173 |
| CGENN* [68] | 0.942 | 0.9869 | 500 | 2172 |
| PELICAN* [10] | **0.9426** ± 0.0002 | **0.9870** ± 0.0001 | – | **2250** ± 75 |
| L-GATr (ours)* | 0.9423 ± 0.0002 | **0.9870** ± 0.0001 | **540** ± 20 | 2240 ± 70 |

Table 1: Top tagging. We compare accuracy, area under the ROC curve (AUC), and inverse background acceptance rate $1/\epsilon_B$ at two different signal acceptance rates (or recall) $\epsilon_S \in (0.3, 0.5)$ for the top tagging dataset from Kasieczka et al. [50]. Lorentz-equivariant methods are indicated with an asterisk*; the best results for each metric are in **bold**. For L-GATr, we show the mean and standard deviation of five random seeds. Baseline results are taken from the literature.

## 4.2 Top tagging

**Problem**  Next, we turn to the problem of classifying whether a spray of reconstructed hadrons originated from the decay of a top quark or any other process. This problem of top tagging is an important filtering step in any analysis that targets the physics of top quarks, the heaviest elementary particle in the Standard Model. Particle collisions involving these particles are of particular interest to physicists because the production and decay probabilities of top quarks are sensitive to several proposed theories of new physics, including for instance the existence of "supersymmetric" particles. We use the established top tagging dataset by Kasieczka et al. [49, 50] as a benchmark and compare to the published results for many algorithms and architectures.

**Results**  As shown in Tbl. 1, L-GATr is on par with or better than even the strongest baselines on this well-studied benchmark.

## 4.3 Generative modelling

**Problem**  Finally, we study the generative modelling of reconstructed events as an end-to-end generation task [17, 18], bypassing the whole simulation chain visualized in Fig. 1. Such generative models can obliterate the computational cost of both the theory computations and the detector simulation at once. However, the high-dimensional distributions of reconstructed particles often have non-trivial kinematic features that are challenging for generative models to learn, for instance the properties of unstable resonances and angular correlations. We focus on the processes $pp \to t\bar{t} + n$ jets, the generation of top pairs with $n = 0 \ldots 4$ additional jets, where the top quarks decay hadronically, $t \to bq'\bar{q}''$.

We train continuous normalizing flows based on an L-GATr network with the Riemannian flow matching objective described in Sec. 3. As baselines, we consider similar flow matching models, but use MLP and Transformer networks as score models, as proposed by Refs. [18, 45]. We also construct a flow matching model using the E(3)-equivariant GATr from Ref. [14]. Finally, we also train JetGPT [18] model, an autoregressive transformer architecture developed for particle physics that is not equivariant to the Lorentz symmetry.

**Kinematic distributions**  We begin with a qualitative analysis of the samples from the generative models. In Fig. 4 we show example marginal distributions from the different models and compare them to the ground-truth distribution in the test set. We select three marginals that are notoriously difficult to model correctly for generative models. While the differences are subtle and only visible in tails and edges of the distributions, L-GATr matches the true distribution better than the baselines. However, none of the models are able to capture the kinematics of the top mass peak at percent-level precision yet.

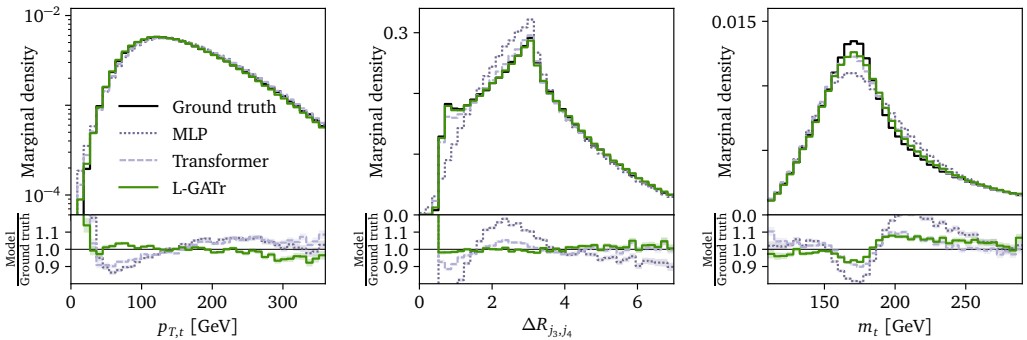

Figure 4: Generative modelling: Marginal distributions of reconstructed particles in the $pp \to t\bar{t} + 4$ jets process. We compare the ground-truth distribution (black) to three generative models: continuous normalizing flows based on a Transformer, MLP, or our L-GATr network. The three marginals shown represent kinematic features that are known to be challenging. The L-GATr flow describes them most accurately.

**Log likelihood** Next, we evaluate the generative models quantitatively through the log likelihood of data samples under the trained models; see Appendix C for details. The left panel of Fig. 5 shows that the L-GATr models outperform all baselines across all different jet multiplicities. They maintain this performance advantage also for smaller training data size, as shown in the right panel. The flow models, including L-GATr, are more data-efficient than the autoregressive transformer JetGPT.

**Classifier two-sample test** How close to the ground-truth distribution are these generative models really? Neither marginal distributions nor log likelihood scores fully answer this question, as the former neglect most of the high-dimensional information and the latter do not have a known ground-truth value to compare to. We therefore perform a classifier two-sample test [55]. We find that L-GATr samples are difficult to distinguish from the ground-truth distribution: a classifier trained to discriminate them achieves only a ROC AUC of between 0.51 and 0.56, depending on the process. In contrast, Transformer and MLP distributions are more easily discriminated from the background, with ROC AUC results between 0.58 and 0.85. For details, see Appendix C.

**Effect of Riemannian flow matching** How important was our choice of probability paths through Riemannian flow matching for the performance of these models? In Tbl. 2 we compare the log likelihood of CFM L-GATr models that differ only in the probability paths. Clearly, the Riemannian flow matching approach that allows us to encode geometric constraints is crucial for a good performance. We find similarly large gains for all architectures.

| Probability paths | NLL |
|---|---|
| Euclidean | -30.11 $\pm$ 0.98 |
| RFM | **-32.65** $\pm$ 0.01 |

Table 2: Benefit of Riemannian flow matching for generative models. We show the negative log likelihood on the $t\bar{t} + 0j$ test set (lower is better).

### 4.4 Computational cost and scalability

Finally, we briefly comment on L-GATr's computational cost. Compared to a vanilla Transformer, the architecture has some computational overhead because of the more complex linear maps. However, it scales exactly in the same way to large particle multiplicities, where both architectures are bottlenecked by the same dot-product attention mechanism. At the same time, L-GATr is substantially more efficient than equivariant architectures based on message passing, both in terms of compute and memory. This is because high-energy physics problems do not lend themselves to sparse graphs, and for dense graphs, dot-product attention is much more efficient. See Appendix C for our measurements.

## 5 Discussion

Out of all areas of science, high-energy physics is a strong contender for the field in which symmetries play the most central role. Surprisingly, while particle physicists were quick to embrace machine learning, architectures tailored to the symmetries inherent in particle physics problems have received comparably little attention.

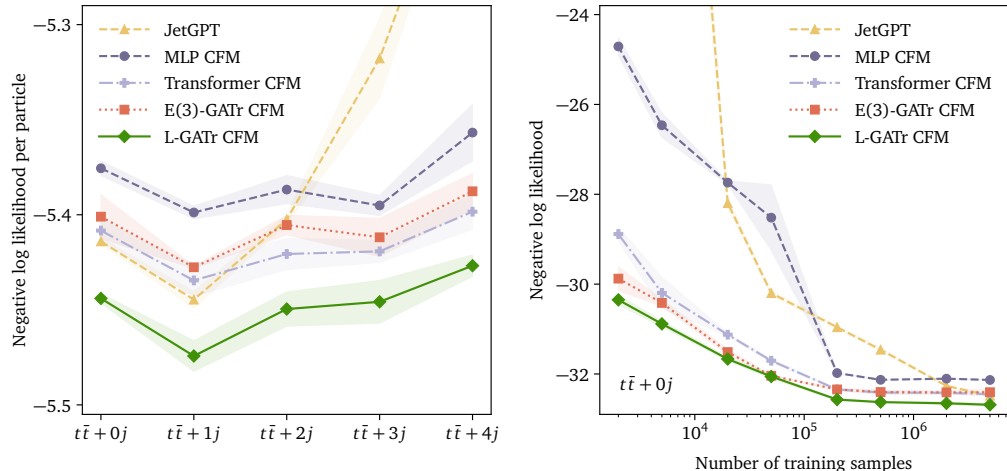

Figure 5: Generative modelling: negative log likelihood on the test set (lower is better). **Left**: For different processes. **Right**: As a function of the training dataset size. We show the mean and standard deviation of three random seeds. The L-GATr conditional flow matching (CFM) model outperforms all other CFM models as well as the autoregressive transformer JetGPT, across all processes and all training set sizes.

We introduced the Lorentz Geometric Algebra Transformer (L-GATr), a versatile architecture with strong inductive biases for high-energy physics: its representations are based on particle four-momenta, extended to higher orders in a geometric algebra, and its layers are equivariant with respect to the Lorentz symmetry of special relativity. At the same time, L-GATr is a Transformer, and scales favorably to large capacity and large numbers of input tokens.

We demonstrated L-GATr's versatility on diverse regression, classification, and generative modelling tasks from the LHC analysis workflow. For the latter, we constructed the first Lorentz-equivariant generative model based on Riemannian flow matching. Across all experiments, L-GATr performed as well as or better than strong baselines.

Still, L-GATr has its limitations. While the architecture scales better than comparable message-passing networks, it has some computational overhead compared to, for instance, efficient Transformer implementations. And while L-GATr should in principle be suitable for pretraining across multiple problems, we have not yet investigated its potential as a foundation model.

While the LHC is preparing for the high-luminosity runs and its legacy measurements, the high-energy physics community is optimizing all steps of the analysis pipeline. Deploying performant and data-efficient architectures such as L-GATr could improve this pipeline in many places. We hope that this will ultimately contribute to more precise measurements of nature at its most fundamental level.

**Acknowledgements**

We would like to thank Taco Cohen and Anja Butter for fruitful discussions, Nathan Hütsch for help with conditional flow matching, and David Ruhe for support with the CGENN code.

J.S., V.B. and T.P. are supported by the Baden-Württemberg-Stiftung through the program *Internationale Spitzenforschung*, project *Uncertainties — Teaching AI its Limits* (BWST_IF2020-010), the Deutsche Forschungsgemeinschaft (DFG, German Research Foundation) under grant 396021762 – TRR 257 *Particle Physics Phenomenology after the Higgs Discovery*, and through Germany's Excellence Strategy EXC 2181/1 – 390900948 (the *Heidelberg STRUCTURES Excellence Cluster*). J.S. is funded by the Carl-Zeiss-Stiftung through the project *Model-Based AI: Physical Models and Deep Learning for Imaging and Cancer Treatment*. V.B. is supported by the BMBF Junior Group *Generative Precision Networks for Particle Physics* (DLR 01IS22079). V.B. acknowledges financial support from the Grant No. ASFAE/2022/009 (Generalitat Valenciana and MCIN, NextGenerationEU PRTR-C17.I01). J.T. is supported by the National Science Foundation under Cooperative Agreement PHY-2019786 (The NSF AI Institute for Artificial Intelligence and Fundamental Interactions, http://iaifi.org/), by the U.S. Department of Energy Office of High Energy Physics under grant number DE-SC0012567, and by the Simons Foundation through Investigator grant 929241.

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

# A  Geometric algebra

Geometric algebras are mathematical objects that were initially used for physics. Although they have been used in machine learning for decades [9], they have seen a recent uptick in popularity [11, 14, 38, 57, 68, 69]. In this section, we will introduce geometric algebras and the relevant concepts.

An algebra is a vector space that is equipped with an associative bilinear product. Given a vector space $V$ with a symmetric bilinear inner product, we can construct an algebra $\mathbb{G}(V)$, called the geometric or Clifford algebra, in the following way: choose an orthogonal basis $e_i$ of the original $d$-dimensional vector space $V$. Then, the algebra has $2^d$ dimensions with a basis given by elements $e_{j_1} e_{j_2} ... e_{j_k} =: e_{j_1 j_2 ... j_k}$, with $1 \leq j_1 < j_2 < ... < j_k \leq d, 0 \leq k \leq d$. For example, for $V = \mathbb{R}^3$, with orthonormal basis $e_1, e_2, e_3$, a basis for the algebra $\mathbb{G}(R^3)$ is

$$1, e_1, e_2, e_3, e_{12}, e_{13}, e_{23}, e_{123} \,. \tag{5}$$

An algebra element spanned by basis elements with $k$ indices is called a $k$-vector or a vector of *grade* $k$. A generic element whose basis elements can have varying grades is called a *multivector*. A multivector $x$ can be projected to a $k$-vector with the grade projection $\langle x \rangle_k$.

The product on the algebra, called the geometric product, is defined to satisfy $e_i e_j = -e_j e_i$ if $i \neq j$ and $e_i e_i = \langle e_i, e_i \rangle$, which by bilinearity and associativity fully specifies the algebra. Given an algebra $\mathbb{G}(V)$, there is a group $\mathrm{Pin}(V)$ that is generated by the 1-vectors in the algebra with norm $\pm 1$, and whose group product is the geometric product. This group has a linear action $\rho : \mathrm{Pin}(V) \times \mathbb{G}(V) \to \mathbb{G}(V)$ on the algebra defined such that for any unit 1-vector $u \in \mathrm{Pin}(V)$ and 1-vector $x \in \mathbb{G}(V)$

$$\rho(u, x) = -uxu^{-1}. \tag{6}$$

The action is defined to be an algebra homomorphism, meaning that for any $u \in \mathrm{Pin}(V), x, y \in \mathbb{G}(V), \rho(u, xy) = \rho(u, x)\rho(u, y)$. Also, it is a group action, meaning that for any two group elements $u, v \in \mathrm{Pin}(V), \rho(uv, x) = \rho(u, \rho(v, x))$. As the group $\mathrm{Pin}(V)$ is generated by products of 1-vectors, and the algebra $\mathbb{G}(V)$ is generated by linear combinations and geometric products, this fully specifies the action $\rho$.

**Space-time geometric algebra**   In this paper, we use the geometric algebra $\mathbb{G}_{1,3} = \mathbb{G}(\mathbb{R}^{1,3})$ based on four-dimensional Minkowski space $\mathbb{R}^{1,3}$, which has an orthogonal basis with one basis vector $e_0$ satisfying $\langle e_0, e_0 \rangle = +1$ and for $i = 1, 2, 3$ a basis vector $e_i$ satisfying $\langle e_i, e_i \rangle = -1$. The Pin group $\mathrm{Pin}(\mathbb{R}^{1,3})$ is a double cover of the Lorentz group $\mathrm{O}(1, 3)$. As we do not require equivariance to time reversals or spatial mirrorings, we are only interested in equivariance to the connected subgroup $\mathrm{SO}^+(1, 3)$.

**Equivariance**   The fact that $\rho$ is an algebra homomorphism is equivalent to saying that the geometric product is equivariant to $\mathrm{Pin}(V)$. Furthermore, the grades in a geometric algebra form subrepresentations [14, Prop. 2]. Thus, the grade projections are equivariant. The pseudoscalar is a one-dimensional real representation, and thus must be invariant to any connected subgroup of $\mathrm{Pin}(V)$. Therefore, multiplying by the pseudoscalar is equivariant to the connected group $\mathrm{SO}^+(1, 3)$. Hence, the linear layer in Eq. (1) is equivariant, for any value of the parameters $v, w$.

To show that this forms a complete basis of all equivariant linear maps, we use the numerical approach of De Haan et al. [36], based on Finzi et al. [40]. Numerically, we find a 10-dimensional space of equivariant maps, indicating the basis in Eq. (1) is complete, which proves Prop. 1.

**Expressivity**   A fortiori, De Haan et al. [36] showed that for several geometric algebras, any equivariant map $\mathbb{G}(V) \times ... \times \mathbb{G}(V) \to \mathbb{G}(V)$ that is a polynomial function of the coefficients of $\mathbb{G}(V)$, can be expressed as linear combinations of grade projections, geometric products and invariant multivectors, and is thus expressible by GATr. This argument holds for any geometric algebra that is based on a vector space $V$ with a non-degenerate metric, which is the case for the Minkowski space $\mathbb{R}^{1,3}$. Hence, this expressivity argument can be extended to L-GATr: the operations in its MLPs are able to express any polyomial map of $\mathbb{G}_{1,3}$ mutlivectors.

# B  Architectures

**Lorentz Geometric Algebra Transformer (L-GATr)**   Our main contribution is the L-GATr architecture, described in detail in Sec. 3. Our implementation is in part based on the Geometric Algebra

Transformer [14] code in version 1.0.0.[9] Unlike [14], we use multi-head attention, not multi-query attention.

**Clifford group equivariant neural network (CGENN)**   We use the CGENN architecture [68] as a baseline. We use the official implementation[10] and adapt their top-tagging code to our amplitude regression experiments.

**Deep Sets with invariants (DSI)**   We build a new architecture based on the Deep Sets framework [77] that uses momentum invariants as part of the input for tackling the amplitude surrogate task. Deep Sets is a permutation-invariant architecture that applies the same function to each element of an input set, aggregates the results with a permutation-invariant operation like a sum, and processes the outputs with another function.

Our adaptation for the amplitude regression tasks applies the Deep Sets approach to each subset of identical particles in the inputs, as the amplitudes are manifestly invariant under permutations of the four-momenta of particles of the same type. We thus apply a different preprocessing to each particle type and aggregate them separately. In addition to the particle-specific latent space samples, the input to our main network also includes the momentum invariants for all the particles involved in the process. DSI thus combines a Lorentz-equivariant, permutation-equivariant path with a non-Lorentz-equivariant non-permutation-equivariant path, allowing the network to learn whether to rely on equivariant or non-equivariant features.

Both preprocessing units and the main network are implemented as MLPs with GELU nonlinearities. Using this model, we are able to obtain optimal performance for simple interactions, but we observe a poor scaling behavior for prediction quality as we increase particle multiplicity.

**Geometric Algebra Perceptron (GAP)**   To ablate to what extent L-GATr's performance is due to the geometric algebra representations and its equivariance properties and to what extent due to the Transformer architecture, we use L-GATr's MLP block as a standalone network. We call it "Geometric Algebra Perceptron" (GAP). Instead of structuring our data as a set of tokens, we format the particle data as a single list of channels. This implies that interactions between particles within the model will be carried out in the linear layers, as opposed to the attention mechanism we use in L-GATr.

**Transformer**   Orthogonally to GAP, we also use a vanilla Transformer as a baseline in our experiments. We use a pre-LayerNorm Transformer with multi-head attention and GELU nonlinearities. This setup mirrors that of L-GATr as closely as possible.

**Multilayer perceptron (MLP)**   The MLP represents our simplest baseline, formulated as a stack of linear layers with GELU nonlinearities.

## C   Experiment details

### C.1   Surrogates for QFT amplitudes

**Dataset**   We generate training and evaluation data consisting of phase space inputs and their corresponding interaction amplitudes for processes $q\bar{q} \to Z + ng$, $n = \{1, 2, 3, 4\}$, where an initial quark-antiquark pair interact to produce a $Z$ boson and a variable number of gluons. The amplitudes are invariant under the permutation of identical particles and under Lorentz transformations. These datasets are generated by the MadGraph Monte Carlo event generator [4] in two steps.[11] First, we use a standard run to generate the phase space distributions. This standard run applies importance sampling to produce unweighted samples, that is, events that are distributed according to the probability distribution that describes the physical interactions. Second, we re-compute the amplitude values corresponding to these phase-space samples with MadGraph's standalone module.

---

[9]Available at `https://github.com/Qualcomm-AI-research/geometric-algebra-transformer` under a BSD-3-Clause-Clear license.

[10]Available at `https://github.com/DavidRuhe/clifford-group-equivariant-neural-networks` under a MIT license.

[11]Available at `https://launchpad.net/mg5amcnlo` under a UoI-NCSA open source license.

We produce four datasets, each with a different number of gluons. Each dataset consists of $4 \times 10^5$ samples for training, $10^5$ for validation, and $5 \times 10^5$ for testing. Our event sets feature kinematic cuts in the transverse momentum of the outgoing particles ($p_T > 20$ GeV) and on the angular distance between the gluons ($\Delta R = \sqrt{\Delta \eta^2 + \Delta \phi^2} > 0.4$).

For the learning problem, we affinely normalize the amplitudes $y$ to zero mean and variance one:

$$\hat{y}_i = \frac{\log(y_i) - \overline{\log(y_i)}}{\sigma_{\log(y_i)}} . \tag{7}$$

**Models**  For the L-GATr model, we embed each particle as a token. It is characterized with its four-momentum, embedded as a grade-1 multivector in the geometric algebra, as well as a one-hot embedding of the particle type, embedded as a scalar. We standardize the four-momentum inputs to unit variance, using the same normalization for each component to preserve Lorentz equivariance. In addition to the particle tokens we use one "global" token, initialized to zero. After processing these inputs with an L-GATr network, we select the scalar component of the global token and identify it as the amplitude output. We use 8 attention blocks, 32 multivector and 32 scalar channels, and 8 attention heads, resulting in $1.8 \times 10^6$ learnable parameters.

For the CGENN, we minimally alter the graph neural network version of the model built for the top tagging classification task so that it is able to perform amplitude regression. We keep the hyperparameters proposed by [68] and use 72 hidden node features, 8 hidden edge features, and 4 blocks. This model features around $3.2 \times 10^5$ trainable parameters.

For the GAP we use the same procedure as with L-GATr to embed (and preprocess) the inputs and extract the outputs, the only difference is that the different particles in a given event are distributed as individual channels in the input. This model consists of 8 blocks, 96 multivector and 96 scalar channels, resulting in $2.5 \times 10^6$ learnable parameters.

For the Transformer baseline, we again include particle tokens through a one-hot embedding. In this case the inputs $x$ are preprocessed by performing standarization, defined as

$$\hat{x}_i = \frac{x_i - \overline{x}_i}{\sigma_{x_i}}, \tag{8}$$

where the mean and the standard deviation are computed over each particle input separately. As for the network structure, we use 8 attention blocks, 128 hidden channels and 8 attention heads, resulting in $1.3 \times 10^6$ learnable parameters.

For the DSI, we implement input standarization in the same way we do with the Transformer, but we also apply the same transformation to each of the momentum invariant inputs separately. As for the layer structure, all MLP modules have 4 layers with 128 hidden channels each, and we set up the preprocessing units so that they output 64-dimensional latent space samples. All in all, we end up with $2.6 \times 10^5$ parameters in total.

For the MLP, we once again apply standarization, this time over the whole input. The network consists of 5 layers and 128 hidden channels amounting to $7 \times 10^4$ learnable parameters.

**Training**  All models are trained by minimizing a mean squared error (MSE) loss on the preprocessed amplitude targets and by making use of the Adam optimizer. We use a batch size of 256 and a fixed learning rate of $10^{-4}$ for all baselines. As for the number of training steps, MLP and DSI are trained for around $2.5 \times 10^6$ iterations, the Transformer for around $10^6$ iterations and GAP, CGENN and GATr for $2.5 \times 10^5$ iterations. We use no regularization method for any of our baselines. We use early stopping across all training runs.

## C.2   Top tagging

**Dataset**  We use the reference top quark tagging dataset by Kasieczka et al. [49, 50].[12] The data samples are structured as point clouds, with each event simulating a measurement by the ATLAS experiment at detector level. Signal samples originate from the decay of a top quark, while the rest of the events are generated by standard background processes. The dataset consists of $1.2 \times 10^6$ events for training and $4 \times 10^5$ each for validation and testing.

---

[12]Available at https://zenodo.org/records/2603256 under a CC-BY 4.0 license.

**Models**    As the top-tagging datasets operate with reconstructed particles as measured by a detector, we include the proton beam direction as an extra particle input to the network, which partially breaks the Lorentz equivariance of the process. We encode it as a rank-2 multivector representing the plane orthogonal to the beam direction. We also add another token containing the time direction $(1, 0, 0, 0)$ as a rank-1 multivector. The time direction is required to break the special orthochronous Lorentz group $SO^+(1, 3)$ down to the subgroup $SO(3)$.

Otherwise, we use the same setup as in the amplitude regression task. We use 12 attention blocks, 16 multivector and 32 scalar channels and 8 attention heads, resulting in $1.1 \times 10^6$ learnable parameters.

**Training**    L-GATr is trained by minimizing a binary cross entropy (BCE) loss on the top quark labels. We train it for $2 \times 10^5$ steps using the EvoLved Sign Momentum (LION) [27] optimizer with a weight decay of 0.2 and a batch size of 128. We use a Cosine Annealing scheduler [56] with the maximum learning rate set at $3 \times 10^{-4}$.

## C.3    Generative modelling

**Dataset**    The $t\bar{t} + n$ jets, $n = 0...4$ dataset is simulated with the MadGraph 3.5.1 event generation toolchain, consisting of MadEvent [4] for the underlying hard process, Pythia 8 [72] for the parton shower, Delphes 3 [35] for a fast detector simulation, and the anti-$k_T$ jet reconstruction algorithm [21] with $R = 0.4$ as implemented in FASTJET [22]. The Pythia simulation does not include multi-parton interactions. We use the ATLAS detector card for the Delphes detector simulation, apply the phase space cuts $p_T > 22$ GeV, $|\eta| < 5, \Delta R = \sqrt{\Delta\phi^2 + \Delta\eta^2} < 0.5$ and require 2 $b$-tagged jets. The events are reconstructed with a $\chi^2$-based reconstruction algorithm [2], and identical particles are ordered by $p_T$.

The sizes of the $t\bar{t} + n$ jets, $n = 0...4$ datasets reflect the frequency of the respective processes, resulting in $9.8 \times 10^6$ ($n = 0$), $7.2 \times 10^6$ ($n = 1$), $3.7 \times 10^6$ ($n = 2$), $1.5 \times 10^6$ ($n = 3$) and $4.8 \times 10^5$ ($n = 4$) events. On each dataset, 1% of the events are set aside as validation and test split. We rescale the four-momenta $p$ by the standard deviation of all five datasets 206.6 GeV for processing with neural networks.

**Models**    The L-GATr score network operates in Minkowski space $p = (E, p_x, p_y, p_z)$, whereas flow matching happens in the physically motivated coordinates $y = (y_m, y_p, \eta, \phi)$ defined in Eq. (4). After transforming $y$ into $p$, we embed each particle $p$ into geometric algebra representations. We use scalar channels for the one-hot-encoded particle type and the flow time, for which we use a Gaussian Fourier Projection with 8 channels [73]. We add the same symmetry breaking inputs that we used for the top tagging task, this time as extra multivector channels. The output of the L-GATr network is a vector field in Minkowski space $(v_E, v_{p_x}, v_{p_y}, v_{p_z})$. To obtain vector fields in the manifold coordinates $y$, we multiply with the Jacobians of the transformation $p \to y$ to obtain $(v_{y_m}, v_{y_p}, v_\eta, v_\phi)$. Because of the logarithm in the change of variables, the Jacobians for $y_m, y_p$ can take on large values, which makes training unstable. To avoid this complication, we extract $\tilde{v}_{y_m}, \tilde{v}_{y_p}$ directly from the scalar output channels of L-GATr and use these values as vector field $v_{y_m}, v_{y_p}$. Like the symmetry-breaking inputs, this procedure breaks the Lorentz symmetry down to the residual symmetry group of the measurement process. We find it beneficial to perform the transformation $y \leftrightarrow p$ at 64-bit floating-point precision, which has no noticable effect on the computational cost. We use an L-GATr network with 16 multivector channels, 32 scalar channels, 6 L-GATr blocks, and 8 attention heads, totalling $5.4 \times 10^5$ learnable parameters.

We implement the E(3)-GATr score network following the same prescription outlined above for L-GATr. The only difference is that we only transform between $y$ and $(y_m, p_x, p_y, p_z)$ to avoid the large jacobians from the $y_m \leftrightarrow m$ transformation. Our task only requires SO(3)-equivariance, therefore we do not use of the translation-equivariant input and output representations of E(3)-GATr, but we keep the internal translation-equivariant representations. For each particle, we embed the three-momentum $(p_x, p_y, p_z)$ into the vector component of the multivector, and $y_m$ as a scalar.

The Transformer score network directly operates in the physically motivated $y$ space. Each particle is embedded into one token, with 4 channels for the components of $y$, 8 channels for the flow time in a Gaussian Fourier Projection, and the one-hot-encoded particle type. The network has 108 channels, 6 Transformer blocks, and 8 attention heads, totalling to $5.7 \times 10^5$ learnable parameters.

The MLP score network also operates in $y$ space. Each event is embedded as a list of the components of $y$, together with the time embedded with Gaussian Fourier Projection using 8 channels. The network has 336 channels and 6 blocks, with $5.9 \times 10^5$ learnable parameters.

We implement the JetGPT model closely following Ref. [18]. JetGPT is an autoregressive mixture model, i.e. its parameters are predicted autoregressively for each component of the high-dimensional distribution. We use the same transformer as for the Transformer-CFM to predict the mixture parameters, but using a triangular attention matrix to achieve the autoregressive structure. The components are the same $y$ coordinates that we use for the flow matching models, amounting to four tokens for each particle. We use gaussian mixture models for the non-periodic coordinates $(y_m, y_p, \eta)$, and von Mises mixture models for the periodic coordinate $\phi$. Due to the autoregressive structure, the ordering of components affects the performance. As more conditions are added, the components will be harder to learn using a fixed amount of training data. We order the particles as $(t_1, q_1, q_2, t_2, q_3, q_4, j_1, \ldots j_4)$, and within each particle we order the components as $(\phi, y_p, \eta, y_m)$. We train the model to minimize the joint log-likelihood and sample the components sequentially. Each component is embedded into one token, with 1 channel for the value, and 24 to 40 channels for the one-hot-encoded component type. We use the same transformer as for the Transformer-CFM, with 108 channels, 6 Transformer blocks, and 8 attention heads, but the last layer maps onto the 108 parameters of a mixture model of 36 gaussian or von Mises distributions.

**Training**    All networks are trained for $2 \times 10^5$ iterations with batchsize 2048 using the Adam optimizer with default settings and an initial learning rate of 0.001. We evaluate the validation loss every $10^3$ iterations and decrease the learning rate by a factor of 10 after no improvements for 20 validation steps. We stop training after the validation loss has not improved after 50 validation steps, and pick the network with the best validation loss.

**Base distribution**    The base distribution is defined in the rescaled Minkowski space discussed above. We use standardized Gaussians for the spatial momentum $p_{x,y,z} \sim \mathcal{N}(0,1)$ and the log-transformed squared mass $y_m = \log m^2 \sim \mathcal{N}(0,1)$. We ensure the constraints $p_T > 22$ GeV, $\Delta R > 0.5$ through rejection sampling. We have experimented with other base distributions and find similar performance.

**Probability paths**    The target probability paths for RCFM linearly change the physically motivated coordinates $y = (y_m, y_p, \eta, \phi)$ defined in Eq. (4). We use pseudorapidity $\eta$ instead of true rapidity, since this is easier to implement given the cut on transverse momentum $p_T$. We use a constant diagonal metric, with the squared inverse standard deviation of these coordinates in the training dataset on the diagonal. This is equivalent to a standardization step. We construct periodic target vector fields for angular coordinates $\phi$ by adding factors of $2\pi$ until angular coordinates and angular velocities end up in the interval $[-\pi, \pi]$.

**Negative log-likelihood (NLL) metric**    For any sample $p$ in Minkowski space, we evaluate the log-density of the transformed sample $y$ using the instantaneous change of variables [26]. In other words, we solve the ODE

$$\frac{d}{dt}\begin{pmatrix} x_t \\ f_t(x_t) \end{pmatrix} = \begin{pmatrix} v_t(x_t) \\ -\mathrm{div}(v_t)(x_t) \end{pmatrix} \tag{9}$$

with the initial conditions $x_1 = y$, $f_1(x_1) = 0$. We use the Hutchinson trace estimator to evaluate the divergence of the vector field. Using the base density $P_0$, we then evaluate the density $P_{\mathrm{model}}$ of CFM-generated samples in Minkowski space as

$$-\log P_{\mathrm{model}}(p) = -\log P_0(x_0) + f_0(x_0) + \log \det \frac{\partial p_0}{\partial y_0} + \log \det \frac{\partial y_1}{\partial p_1}. \tag{10}$$

The last two terms are the logarithms of the jacobian determinants for the transformations between Minkowski space and the physically motivated space.

**Classifier two-sample test**    We train a MLP classifier to distinguish generated events from the ground truth. The classifier inputs are full events in the $y$ representation, together with challenging correlations, in particular all pairwise $\Delta R$ values, and the $y$ representations of the reconstructed particles $t, \bar{t}, W^+, W^-$. The classifier network has 256 channels and 3 layers. It is trained for 500 epochs with batchsize 1024, a dropout rate of 0.1, and the Adam optimizer with default settings.

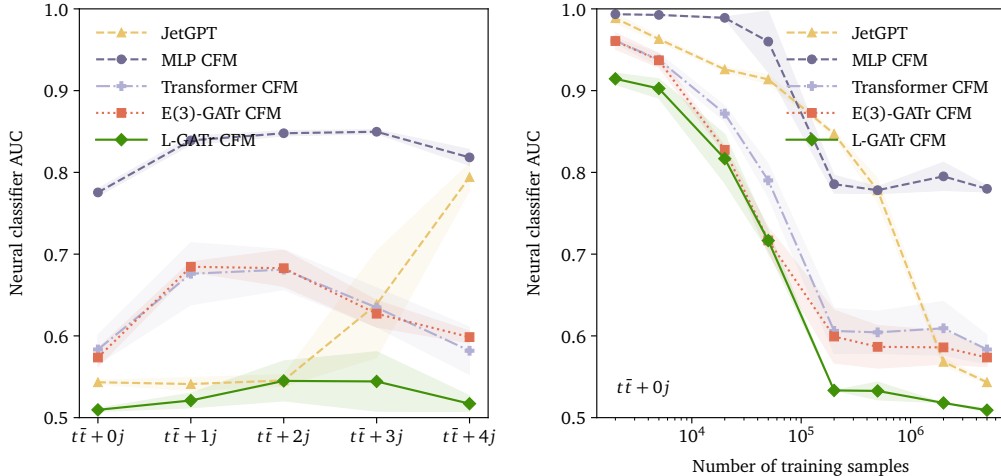

Figure 6: Generative modelling: classifier two-sample tests. We show how well a classifier can discriminate model samples from test samples, measured through the area under the ROC curve (lower is better, 0.5 is ideal). **Left**: For different processes. **Right**: As a function of the training dataset size. We show the mean and standard deviation of three random seeds. The L-GATr flow outperforms the baselines in all processes and all training set sizes.

We start with an initial learning rate of 0.0003 and decrease the learning rate by a factor of 10 after no improvements in the validation loss for 5 epochs. We stop training after 10 epochs without improvements in the validation loss and load the best-validation model afterwards. We use the full ground truth dataset as well as 1M generated events and split into 80% for training and 10% each for testing and validation. We use the AUC of this classifier evaluated on the test dataset as a scalar metric, with the value 0.5 for a perfect generator. Neural classifiers approximate the event-wise likelihood ratio $p_{\text{data}}(x)/p_{\text{model}}(x)$ of single events, which is the most powerful test statistic according to the Neyman-Pearson lemma, and opens many ways to further study the performance of the generator beyond scalar metrics [34].

Our results are shown in Fig. 6. Across training data sizes and processes, the L-GATr flows are more difficult to distinguish from the test samples than the baselines.

## C.4 Computational cost and scalability

In Fig. 7 we compare L-GATr to a message-passing graph neural network (we use CGENN [68]) and a vanilla Transformer in terms of their test-time computational costs. For this comparison, we use small versions of all architectures, consisting of a single model block and around $2 \times 10^5$ learnable parameters. In the case of the Transformer and L-GATr, we fix their layer structure so that inputs going into the attention layer consist of 72 channels. Our measurements are performed with datasets made up by a single sample and all models are run on an H100 GPU.

L-GATr in its current implementation is not yet as efficient as a Transformer for small systems: for up to hundreds of particles, L-GATr takes an order of magnitude longer to evaluate. This is caused by the linear layers in L-GATr, which are more costly to execute than their non-equivariant counterparts and represent a constant computational overhead. However, because L-GATr is based on the same efficient backend for dot-product attention, it scales just like a Transformer to larger systems, and we find the same computational cost for 5000 particles or more.

Compared to an equivariant graph network, L-GATr is clearly more efficient in terms of compute time and memory. Already at small systems, L-GATr can be evaluated an order of magnitude faster. The difference is even more pronounced in terms of memory: the graph network ran out of memory for more than a thousand particles. This is largely because graph network implementations are often optimized for sparse computational graphs, but here we use fully connected graphs: LHC problems often benefit from a fully connected computational graph, because pairwise interactions do not usually decay with the Minkowski norm of the distance between momenta. Transformer-based approaches

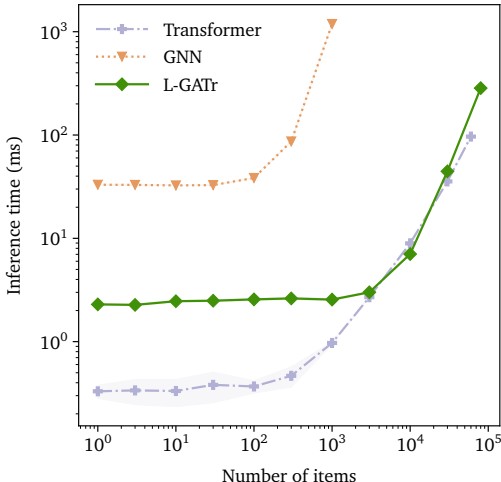

Figure 7: Inference cost (wall-time per forward pass) as a function of the number of particles. We compare L-GATr, a Transformer, and a message-passing graph neural network (we use CGENN [68] but expect similar results for other architectures). The latter runs out of memory when evaluating more than a thousand particles. While we do our best to find comparable settings, such comparisons depend on a lot of choices and should be interpreted with care. Nevertheless, we believe they illustrate that L-GATr scales to large systems like a Transformer, thanks to it being based on dot-product attention.

like L-GATr can thus have substantial computational advantages in particle physics problems that involve a large number of particles.

