# OpenReview forum: "Lorentz-Equivariant Geometric Algebra Transformers for High-Energy Physics"
_NeurIPS.cc/2024/Conference — NeurIPS 2024 poster_

### Official Review · Reviewer_PeE1 · 2024-07-11

**Soundness:** 3
**Presentation:** 3
**Contribution:** 3
**Rating:** 7
**Confidence:** 4

**Summary:**

This paper implements a Lorentz-equivariant transformer and applies it to several problems in particle physics. The main contribution of the paper is not the definition of a conceptually different model. At least it doesn't claim to define a model significantly different from previously proposed models. But it provides a software library (under BSD-3-Clause-Clear license) and a nice set of numerical experiments inspired by physics problems. This is a nice contribution to the community.

**Strengths:**

The paper combines interesting ideas (imposing symmetries arising from physics using invariant theory and clifford algebras) with state-of-the-art machine learning models (transformers, Riemannian flow maching generative modeling). It comes with a software package and multiple experiments where several methods are compared. I see this as a tool that may be used by several members of the AI4science and AI4physics communities.

**Weaknesses:**

The paper is not self-contained. In order to be a valuable tool for our community it would benefit from a more comprehensive explanation of several of its technical aspects.

- The implemented Lorentz equivariant model (what are the inputs?, are they a set of particles? how are they matched to queries, values and keys?)
- The flow-matching approach. Again, what are the inputs, outputs? How's the model defined? What does it produce? The closest thing to an explanation is in page 24, in the context of describing the experiment, but still I don't find this explanation sufficient to understand the model conceptually.
- What are the scientific goals of the experiments defined in the paper? (In the context of fundamental research in high energy physics).

**Questions:**

Please address the questions in the weaknesses section.

Having these clearly answered would improve the paper's clarity, reproducibility, and usability.

I'd be willing to raise my score to 7-8 if all these aspects were clearly explained with concrete explanations and equations, since I believe this can turn into a great tool to the AI 4 physics community.

**Limitations:**

Not applicable.

---

> ### Author Rebuttal · Authors · 2024-08-06
>
> Thank you for the thorough and constructive review. We are particularly happy that you appreciate the open-source release of our library. Thanks as well for the questions and criticisms, which we address in the following.
>
> > The paper is not self-contained.
>
> Thank you for this feedback. Due to the fact that we had three diverse experiments, we had to put many experimental details in the appendix. We will use the extra page in the final version to expand in the main paper our description of architecture, flow-matching approach, and the experiments.
>
> > The implemented Lorentz equivariant model (what are the inputs?, are they a set of particles? how are they matched to queries, values and keys?)
>
> The inputs to our architecture are indeed a set of particles. Each particle is characterized by a) an energy and a momentum, which are concatenated into a "four-momentum" vector, and b) a discrete type; see the beginning of Sec. 2. As we describe in the beginning of Sec. 3.1, the four-vector is embedded into the vector grade of the multivector, and the other grades are zero-padded. The type is represented with a standard one-hot encoding.
>
> This data is then processed with the L-GATr architecture defined in line 177. It consists of a sequence of layers of different types: layer normalization, the construction of keys, queries, and values with linear layers, an attention mechanism, geometric products, gated nonlinearities, and skip connections. We define all these maps on pages 4 and 5.
>
> > The flow-matching approach. Again, what are the inputs, outputs? How's the model defined? What does it produce?
>
> The key component here is a network $v_t(x)$ that takes as inputs a set of particles $x$ as well as a time variable $t$ and outputs one number for each element of the input particle properties. As an example, consider our $t\bar{t}+0j$ experiment, where we generate 6 particles, each characterized by the 4 properties $y_p, y_m, \phi, \eta$ defined in Eq. (4). The L-GATr vector field model embeds them into 6 tokens, each containing a multivector with the 4 properties as well as particle type and symmetry breaking information. The architecture processes this data and outputs a single multivector for each item, and we extract the vector field required for flow matching from the vector component of this multivector. For each of the 6 particles, this vector field contains 4 values corresponding to the 4 properties. For details, see the Models section in Appendix C.3.
>
> To sample with such a model, one first samples initial particle properties from a base distribution (like a multivariate Gaussian in a suitable basis) and calls this $x(1)$. Then one updates the particle properties $x(t)$ by integrating the differential equation $d/dt x(t) = v_t(x)$ from $t = 1$ to $t = 0$ using an off-the-shelf ODE solver. The final values $x(0)$ are the outputs: a set of particle properties sampled from the model. By repeating this process, we can generate different samples of sets of particles.
>
> To train the model, we use the conditional flow matching loss defined in line 122. Intuitively, it trains the network $v_t(x)$ such that the vector field it defines moves particles on shortest paths from the base density to the data density. This is explained in much more detail in the flow-matching paper [1].
>
> > What are the scientific goals of the experiments defined in the paper?
>
> On a high level, all three experiments are part of the data-analysis pipeline at the LHC. We roughly sketch how they fit into this bigger picture in Fig. 1. The goal of any such analysis is either the measurement of parameters of fundamental theories of nature, such as the mass of a particle or the strength of an interaction between particles, or to discover or exclude hypotheses altogether, like in the discovery of the Higgs boson in 2012.
>
> As a concrete example, consider the top-tagging problem of Sec. 4.2. This classification task is used to filter data collected at the Large Hadron Collider for those collisions that are likely to be the result of a top quark decay. Top quarks are the heaviest fundamental particle currently known. We are interested in selecting data that involves their decays because the production and decay probabilities of top quarks depend on whether certain "supersymmetric" particles exist. Analyzing top-decay events will therefore allow us to discover or exclude the existence of supersymmetric particles, but to do that well, we need to be able to filter them precisely, and here we show that L-GATr is a powerful tool for that task.
>
> Now, why do we care about the existence of supersymmetric particles? If they exist, they would help us answer the "hierarchy problem", one of the biggest open questions in fundamental physics. It can be phrased as "Why is the gravitational force so much weaker than the electromagnetic, weak, and strong forces?". The existence of supersymmetric particles would also be relevant for a number of other scientific questions, including the nature of Dark Matter. Apart from supersymmetry, there are more hypotheses for new physics that benefit from confident top-taggers. All in all, if L-GATr can improve the performance on the top tagging task, it may contribute to a better, clearer, or sooner answer to such fundamental physics questions.
>
> We hope that we were able to answer your questions and look forward to discussing further.
>
> **References:**
>
> - [1] Y. Lipman et al, "Flow matching for generative modelling", ICLR 2023

---

> ### Comment · Reviewer_PeE1 · 2024-08-09
> **Thank you for the answers**
>
> I appreciate the explanations. I suggest adding them to the paper or appendix. I increased the score.

---

> > ### Author Response · Authors · 2024-08-12
> >
> > Thank you for the fast response. We are happy to hear you appreciated these explanations, and we'll make sure to add them to the paper.

---

### Official Review · Reviewer_w7CH · 2024-07-12

**Soundness:** 3
**Presentation:** 3
**Contribution:** 3
**Rating:** 6
**Confidence:** 2

**Summary:**

The paper proposes a Lorentz equivariant transformer (L-GATr) based on geometric algebra for high energy physics. It generalizes the Geometric Algebra Transformer (GATr) from $E(3)$ equivariance to the Lorentz group. The proposed transformer is then developed into a generative model based on Riemannian flow matching for particle data. L-GATr is evaluated on several high energy physics tasks, including quantum field theory amplitude surrogates, top tagging, and generative modeling for event reconstruction.

**Strengths:**

1. The paper is well-written and easy to follow. In addition, the problem is well-motivated.
2. The proposed L-GATr generalizes GATr from $E(3)$ to the Lorentz group.
3. As stated by the authors, the Lorentz-equivariant flow matching proposed in section 3.2 is the first generative model proposed for particle physics.
4. Compared to graph-based Lorentz equivariant networks, the transformer architecture is more efficient and scalable.
5. The proposed method is shown to be more data efficient than the baselines in both amplitude surrogates and generative modeling experiments.
6. The ability to scale with the data is also verified in several experiments.
7. The benefits of Riemannian flow matching compared to the Euclidean version are demonstrated in the experiments.

**Weaknesses:**

1. It is a bit unclear to me what has been modified for Lorentz equivariance in the transformer framework. Specifically, (1) and (2) look the same as (4) and (5) in [13]. It’ll be great if the authors can state the changes explicitly.
2. The current presentation of the experiments can be a bit hard to understand for people without a physics background. Adding some basic introduction to the problems can strengthen the paper.
3. In the top tagging experiment, the performance of the proposed method is marginally worse than the baseline method.
4. Although the proposed method is claimed to support symmetry-breaking data, its effect is not well studied in the experiments.

**Questions:**

1. What are $y_m, y_p, \eta$, and $\phi$ in (4)?

**Limitations:**

As mentioned in the paper, the proposed L-GATr has additional computational overhead compared to traditional transformers. Secondly, even though the framework allows additional inputs to address symmetry breaking issues, the effect of such an approach is not well studied. It is unclear how well the proposed method can handle symmetry breaking inputs.

---

> ### Author Rebuttal · Authors · 2024-08-06
>
> Thank you for the thorough and constructive review. We are glad to hear that you appreciated how we generalized the GATr architecture from E(3) to the Lorentz symmetry and the development of the first Lorentz-equivariant architecture. We were particularly happy that you found the paper easy to follow. Thanks as well for the questions and criticisms, which we address one by one.
>
> > It is a bit unclear to me what has been modified for Lorentz equivariance in the transformer framework. Specifically, (1) and (2) look the same as (4) and (5) in [1].
>
> Indeed, the form of these equations and generally the overall architecture design are similar. There are several differences though:
>
> - The multivectors $x$ in our Eq. (1) and Ref. [1]'s Eq. (4) are elements of different geometric algebras with different metrics.
> - The inner product in our Eq. (2) is therefore also different from the one in Ref. [1].
> - The second term in our Eq. (1) and Ref. [1]'s Eq. (4) is different. In both cases, we have a term for the five grade projections, and we can multiply with a multivector that is invariant to the group. For Lorentz equivariance, this invariant multivector is the pseudoscalar $e_{0123}$. In the $E(3)$ case, this invariant multivector is the vector $e_0$.
> - The geometric product used in the MLP is different between both architectures. This is another consequence of using a different metric.
> - L-GATr uses only the geometric product as a bilinear operation in the MLP, while the original GATr had to concatenate that with an equivariant join operation to ensure expressivity. The technical reasons for this are discussed in the appendices of Ref. [1].
> - The layer normalization in Eq. (3) needs to be different from Ref. [1]'s layer normalization. The reason is that the Minkowski norm of special relativity can be negative and the normalization operation has to be robust against that.
>
> > The current presentation of the experiments can be a bit hard to understand for people without a physics background.
>
> Thank you for this feedback. It is important to us that both machine learners and particle physicists can understand the paper. We will use the extra page in the final version to substantially expand the introduction to particle physics and to the concrete experiments.
>
> > In the top tagging experiment, the performance of the proposed method is marginally worse than the baseline method.
>
> It's true we do not outperform the strongest baselines on this task (unlike on the other two tasks). We consider our performance and that of the strongest models in the literature to be on par for all practical purposes. In terms of accuracy, the strongest model and L-GATr differ only by 0.09 percentage points, and on one of the four established metrics L-GATr achieves the best score to date.
>
> > Although the proposed method is claimed to support symmetry-breaking data, its effect is not well studied in the experiments.
>
> Actually, in two of our three experiments we use symmetry-breaking data: the direction of the particle beam is provided as an input in the top-tagging and generative modelling problems. This is important, because the detector measurements depend on this direction. (In contrast, the amplitude regression task does not include detector effects, so additional symmetry-breaking inputs are neither needed nor provided there.)
>
> To test the relevance of these symmetry-breaking inputs, we re-ran the top-tagging experiment for a version of L-GATr *without* the beam direction as an input. We found that this decreased the model's performance as follows:
>
> Model | Accuracy | AUC | $1/\epsilon_B$ ($\epsilon_S = 0.5$) | $1/\epsilon_B$ ($\epsilon_S = 0.3$)
> --- | --- | --- | --- | ---
> L-GATr (original) | 0.9417 | 0.9868 | 548 | 2148
> L-GATr (no beam direction) | 0.9403 | 0.9860 | 440 | 1840
>
> The beam direction is thus important to analyze data that involve detector effects.
>
> > What are $y_m$, $y_p$, $\eta$, and $\phi$ in (4)?
>
> These four variables form an alternative basis for the particle four-momenta that is better aligned with the physically relevant properties of a particle in the context of a collider experiment: $\eta$ and $\phi$ represent the angle in which a particle is moving, $y_p$ is a measure of the speed with which it moves away from the collision, and $y_m$ is related to its mass. A convenient property for flow matching is that the support of the distribution is convex in this space. The dictionary that maps $(y_m, y_p, \eta, \phi)$ to four-momentum $(E, p_x, p_y, p_z)$ is given in Eq. (4) and just after it. We will add the inverse map from four-momenta to this alternative basis and improve the discussion.
>
> > the proposed L-GATr has additional computational overhead compared to traditional transformers.
>
> It's true that L-GATr has some computational overhead over a standard transformer architecture, but we would like to stress that it is managable for almost all particle-physics applications.
>
> In Appendix C.4 of our paper we show timing measurements. A forward pass through the tested model takes between 2ms and 100ms when up to 40k particles are processed. This is fast enough for almost all steps of the data-analysis pipeline and in fact negligible compared to the simulators. (There are a few exceptions that require faster inference, most notably the trigger, but these specialized cases go beyond the scope of the paper.)
>
> Compared to a standard transformer, L-GATr is slower for small numbers of particles. However, it is comparably fast when the number of particles is large, since then the attention operation is the bottleneck for both. Compared to a message-passing network, a popular type of equivariant architecture, L-GATr is consistently faster and scales much better to large numbers of particles.
>
> We hope that we were able to answer your questions and look forward to discussing further.
>
> **References**:
>
> - [1] J. Brehmer et al, "Geometric Algebra Transformer", NeurIPS 2023

---

> > ### Comment · Reviewer_w7CH · 2024-08-13
> >
> > Thank you for explaining the differences between the proposed work and [1]. It's clear to me now the difference between the two. It is also great to hear the authors would provide some background materials for people without physics backgrounds.
> >
> > However, some of the weaknesses remain. The beam direction doesn't seem to significantly impact the network performance in the top-tagging experiment (0.9403 v.s. 0.9417). As a result, I'm still not convinced how well the network can deal with actual symmetry-breaking scenes and to what extent the network can handle symmetry-breaking. Therefore, I keep my rating unchanged.

---

> ### Author Response · Authors · 2024-08-14
>
> Thank you for the response. We are glad we were able to explain the novelties in our architecture better.
>
> As for the effect of symmetry breaking, we tend to disagree that the difference between an accuracy of 0.9403 and 0.9417 is insignificant on this benchmark. This difference is similarly big to the gap between equivariant networks and the top-performing non-equivariant networks (ParticleNet, ParT) outlined in Table 1.
>
> To make the relevance of symmetry-breaking inputs clearer, we also re-ran the generative modelling experiment without them. Because of limited time, we ran shorter experiments (50% of the training time quoted in the paper). This is what we find:
>
> Model | NLL (↓) | AUC (↓)
> --- | --- | ---
> L-GATr (original) | **-32.5** | **0.59**
> L-GATr (no beam direction) | -18.4 | 0.99
>
> So symmetry-breaking inputs are crucial for this experiment. This stark difference is not surprising. Consider the AUC metric, in which we train a classifier to distinguish between samples generated by the model and test data. For a generative model with unbroken symmetry, the distribution is invariant under rotations: it generates samples with particles moving in any direction with equal likelihood. This is very different from the data distribution, which is not invariant under rotations: the beam direction leads to a strongly preferred direction for the produced particles. It is easy for a classifier to spot this difference, leading to the high AUC. A similar argument can be made for the log likelihood.

---

### Official Review · Reviewer_srsp · 2024-07-13

**Soundness:** 3
**Presentation:** 4
**Contribution:** 4
**Rating:** 6
**Confidence:** 3

**Summary:**

The paper proposes the Lorentz Geometric Algebra Transformer (L-GATr) for high-energy physics tasks. This model extends the Geometric Algebra Transformer by incorporating relativistic considerations. Specifically, L-GATr supports partial and approximate symmetry for symmetry-breaking inputs and is applied to generative models.

**Strengths:**

- The motivation of the paper is strong, addressing an important application of equivariance models. While symmetries are prevalent in high-energy physics, there are few applications of equivariance models in this field.
- The paper clearly distinguishes their work from existing research, emphasizing the significance of their contributions.
- They connect the proposed model to the generative framework, making it potentially applicable to a broader range of tasks.

**Weaknesses:**

- The model's performance in Table 1 is not optimal.
- The experiments on generative modeling lack comparisons with other equivariant models or SOTA models.
- The scalability of the model remains limited, which reduces its suitability for high-energy physics applications.

**Questions:**

I think another interesting comparison would be to evaluate the proposed model against traditional sampling methods, especially if the goal is to design machine learning models that are more efficient than traditional methods (within an acceptable error tolerance). Could the paper include this comparison? I believe a positive result could attract attention from researchers in the high-energy physics community.

**Limitations:**

The efficacy and scalability of the proposed framework may not yet suffice to replace traditional methods.

---

> ### Author Rebuttal · Authors · 2024-08-06
>
> Thank you for the thorough and constructive review. We are glad that you liked the motivation of our work, found our contributions significant, and appreciated the generative modelling part. Thanks as well for the questions and criticisms, which we address in the following.
>
> > The model's performance in Table 1 is not optimal
>
> It's true we do not outperform the strongest baselines on this task (unlike on the other two tasks). We consider our performance and that of the strongest models in the literature to be on par for all practical purposes. In terms of accuracy, the strongest model and L-GATr differ only by 0.09 percentage points, and on one of the four established metrics L-GATr achieves the best score to date.
>
> > The experiments on generative modeling lack comparisons with other equivariant models or SOTA models.
>
> We are not aware of any other Lorentz-equivariant generative models. If you know of any, we would appreciate if you could point us to them; we would be eager to add them to the comparison.
>
> The transformer-based flow-matching model that we compare to is the strongest baseline we could come up with. Flow matching is establishing itself as a standard in high-energy physics. For instance, Buhmann et al. [1] compare the performance of GAN, diffusion, and flow-matching models based on the same architectural elements and find flow matching to work best.
>
> That being said, we agree that we should compare to more diverse baselines, especially those with open-source implementations. We started experiments with SO(3)-equivariant architectures and non-equivariant autoregressive transformers [2] and will include them in the final version. Thank you for the suggestion!
>
> > The scalability of the model remains limited. [...] The efficacy and scalability of the proposed framework may not yet suffice to replace traditional methods.
>
> Could you elaborate what you mean here? In terms of scaling the model size, we used L-GATr models with millions of parameters without issues.
>
> In terms of scaling to large training data sets, we see in Fig. 3 that L-GATr benefits from that more than all baselines.
>
> The scaling to large number of particles (tokens) is studied in Appendix C.4. A forward pass through the tested L-GATr model takes between 2ms and 100ms when up to 40k particles are processed. That is fast enough for almost all steps of the particle-physics data-analysis pipeline and in fact negligible compared to the simulators. (There are a few exceptions that require faster inference, most notably the trigger, but these specialized cases go beyond the scope of the paper.)
>
> > evaluate the proposed model against traditional sampling methods
>
> If we understand this suggestion correctly, we are already doing this: we evaluate our generative model by comparing samples from it to samples from the simulator, which are based on traditional Monte-Carlo sampling. We quantify the difference between these two distributions by comparing marginal distributions, evaluating the log likelihood, and through a classifier two-sample test. (Unfortunately, the density of the traditional samples is not tractable, which makes it difficult to define more objective metrics for the high-dimensional samples.)
>
> To the extent that these metrics can measure it, we find that the L-GATr model gives us samples that are close to the traditional sampling method, while being orders of magnitude faster to generate: sampling from an L-GATr flow-matching model takes milliseconds per sample, from a fast classical sampler seconds per sample, and from the most accurate state-of-the-art samplers minutes per sample.
>
> We hope that we were able to answer your questions and look forward to discussing further.
>
> **References**
>
> - [1] E. Buhmann et al, "EPiC-ly Fast Particle Cloud Generation with Flow-Matching and Diffusion", arXiv:2310.00049
> - [2] A. Butter et al, "Jet Diffusion versus JetGPT - Modern Networks for the LHC", arXiv:2305.10475

---

### Official Review · Reviewer_PF88 · 2024-07-13

**Soundness:** 3
**Presentation:** 3
**Contribution:** 3
**Rating:** 7
**Confidence:** 4

**Summary:**

The authors propose an architecture for high-energy physics events – the Lorentz Geometric Algebra Transformer, which is equivariant under Lorentz transformations. The architecture is based on the Geometric Algebra Transformer architecture, and generalizes to relativistic scenarios and the Lorentz symmetry. The architecture is demonstrated on regression, classification and generation tasks in particle physics.

**Strengths:**

This article is well-written and effectively communicates its ideas. While the novelty of the research may not be groundbreaking, it offers valuable contributions to the field. The authors conduct a sufficient number of experiments to test their proposed architecture.

**Weaknesses:**

* The proposed architecture, while theoretically appealing, suffers from significant computational overhead due to the addition of Lorentz layers to the already resource-intensive Transformer model.
* The comparison of generative modeling capabilities is limited to flow models, which may not represent the full spectrum of possible approaches.
* The model's performance, while adequate, does not demonstrate a significant improvement over baseline models, which may be a concern given the added computational cost.

**Questions:**

* The experiments are conducted on four vectors of the hard particles. In this case, is it really necessary to employ transformers since the input dimension is not as high? Especially for the regression task, the training set is a relatively small one, yet the model has around 2 million parameters.
* Why is a smaller model used for the top tagging task? Obviously, this task has higher dimensionality and larger datasets compared to the regression task.

**Limitations:**

The authors briefly discussed the limitations in the Discussion section.

---

> ### Author Rebuttal · Authors · 2024-08-06
>
> Thank you for the thorough and constructive review. We are happy to hear that you found our architecture a valuable contribution and the paper well-written. Thanks as well for the questions and criticisms, which we address one by one.
>
> > significant computational overhead
>
> It's true that L-GATr has some computational overhead over a standard transformer architecture, but we find it sufficiently fast for all particle-physics applications we encountered.
>
> In Appendix C.4 of our paper we show timing measurements. A forward pass through the tested model takes between 2ms and 100ms when up to 40k particles are processed. This is fast enough for almost all steps of the data-analysis pipeline and in fact negligible compared to the simulators. (There are a few exceptions that require faster inference, most notably the trigger, but these specialized cases go beyond the scope of the paper.)
>
> Compared to a standard transformer, L-GATr is slower for small numbers of particles. However, it is comparably fast when the number of particles is large, since then the attention operation is the bottleneck for both. Compared to a message-passing network, a popular type of equivariant architecture, L-GATr is consistently faster and scales much better to large numbers of particles. This is consistent with the prior work GATr [1, see Fig. 5 there] from which L-GATr is derived, which shares performance characteristics with L-GATr.
>
> > The comparison of generative modeling capabilities is limited to flow models
>
> True. We initially focused on flow matching baselines because of the stable training, high-quality samples, and the (approximately) tractable likelihood function. Because of these advantages, flow matching is establishing itself as a standard in high-energy physics. For instance, Buhmann et al. [2] compare the performance of GAN, diffusion, and flow-matching models based on the same architectural elements and find flow matching to work best.
>
> Thank you for the suggestion to expand this. We are working on generating additional baseline results for the generative modelling task, including an SO(3)-equivariant architecture and a non-equivariant autoregressive transformer model [3]. We will include the results and discuss them in the final version of the paper.
>
> > The model's performance, while adequate, does not demonstrate a significant improvement over baseline models, which may be a concern given the added computational cost.
>
> In particular in the amplitude regression task, we find L-GATr to consistently outperform the baselines. The advantage over the baselines gets larger for higher-multiplicity final states and more complex interactions. This regime of amplitude modelling is known to be the most challenging for neural surrogates, but is at the same time the most practically important, as theory computations can become prohibitively expensive there [4, 5]. We believe that L-GATr's performance improvements here can have substantial real-world impact.
>
> > Is it really necessary to employ transformers since the input dimension is not as high?
>
> It is not strictly necessary, but transformers lead to the best performance in our experiments. In Fig. 3, for instance, we show that a transformer outperforms a simple MLP on the amplitude regression task. Similarly, our L-GATr outperforms the GAP model, a Lorentz-equivariant MLP we constructed. This is in line with the larger trend in machine learning, which finds transformers to often outperform other models on a large variety of tasks.
>
> > Why is a smaller model used for the top tagging task?
>
> We tuned hyperparameters independently for each of the three problems. We interpret the differences in optimal model size as follows: the top-tagging problem takes many particles as inputs, but the task itself is a comparably simple binary classification problem. In contrast, amplitude regression requires learning complex functions of the particle momenta with high precision, especially for the higher multiplicities; these can be more accurately expressed with a bigger network.
>
> To quantify how important the network capacity is for the amplitude regression problem, we compared our standard L-GATr models with 2 million parameters to a smaller version with 0.7 million parameters:
>
> Model size | MSE ($Z + 1g$) [$10^{-7}$] | MSE ($Z + 4g$) [$10^{-5}$]
> --- | --- | ---
> 2M | 1.91 | 1.48
> 0.7M | 1.62 | 2.71
>
> This confirms that the capacity is more important for the more complex $Z + 4g$ process (though the smaller L-GATr model still outperforms all baselines there).
>
> We hope we were able to address your questions and look forward to discussing further.
>
> **References**
>
> - [1] J. Brehmer et al, "Geometric Algebra Transformer", NeurIPS 2023
> - [2] E. Buhmann et al, "EPiC-ly Fast Particle Cloud Generation with Flow-Matching and Diffusion", arXiv2310.00049
> - [3] A. Butter et al, "Jet Diffusion versus JetGPT - Modern Networks for the LHC", arXiv:2305.10475
> - [4] S. Badger and J. Bullock, "Using neural networks for efficient evaluation of high multiplicity scattering amplitudes", Journal of High Energy Physics 2020
> - [5] S. Badger et al, "Loop amplitudes from precision networks", SciPost Physics 2023

---

### Author Rebuttal · Authors · 2024-08-06

We would like to thank all reviewers for their detailed feedback and questions.

We are excited to read that the reviewers found the work a "valuable contribution to the field" (reviewer **Pf88**), that they appreciated the "significance of [the] contributions" (reviewer **srsp**), and that they found our L-GATr "a tool that may be used by several members of the AI4science and AI4physics communities" (reviewer **PeE1**).

On a technical level, they appreciated that our transformer "generalizes GATr from E(3) to the Lorentz group" (reviewer **w7CH**), that "the Lorentz-equivariant flow matching is the first [such model]" (reviewer **w7CH**), and that we "conduct a sufficient number of experiments to test" it (reviewer **PF88**).
We are particularly encouraged by the reviewers finding that the paper is "well-written and effectively communicates its ideas" (reviewer **srsp**).

&nbsp;

Their critique and questions are helping us improve the paper further. Here we want to highlight three points.

First, reviewers asked about **additional baselines** for the generative modelling experiment.

This is a great suggestion. We started experiments with SO(3)-equivariant architectures and non-equivariant autoregressive transformers [1]. We were not able to finish these in time for this rebuttal, but we will add the results and discuss them for the final version of this paper.

&nbsp;

Second, reviewers asked about the relevance of **symmetry-breaking inputs**.

In two of our three experiments, we break the full Lorentz symmetry by providing the direction of the collider beam pipe as an input to L-GATr. This information is important, both to capture explicit symmetry breaking from the initial beam directions and to model soft breaking of the symmetry from detector effects. In the original paper, we did however not demonstrate this relevance empirically.

We performed a new experiment to measure the importance of symmetry-breaking inputs. We re-ran the top-tagging experiment for a version of L-GATr *without* the beam direction as an input. We found that this decreased the model's performance as follows:

Model | Accuracy | AUC | $1/\epsilon_B$ ($\epsilon_S = 0.5$) | $1/\epsilon_B$ ($\epsilon_S = 0.3$)
--- | --- | --- | --- | ---
L-GATr (original) | 0.9417 | 0.9868 | 548 | 2148
L-GATr (no beam direction) | 0.9403 | 0.9860 | 440 | 1840

L-GATr's ability to break Lorentz symmetry through the inputs is thus relevant in practice.

&nbsp;

Finally, the reviewers were concerned about the **computational overhead and scalability** of L-GATr.

It's true that L-GATr has some computational overhead over a standard transformer architecture, but we find it sufficiently fast for all particle-physics applications we encountered.

In Appendix C.4 of our paper we show timing measurements. A forward pass through the tested model takes between 2ms and 100ms when up to 40k particles are processed. This is fast enough for almost all steps of the data-analysis pipeline of the Large Hadron Collider and in fact negligible compared to the simulators. (There are a few exceptions that require faster inference, most notably the trigger, but these specialized cases go beyond the scope of the paper.)

Compared to a standard transformer, L-GATr is slower for small numbers of particles. However, it is comparably fast when the number of particles is large, since then the attention operation is the bottleneck for both. Compared to a message-passing network, a popular type of equivariant architecture, L-GATr is consistently faster and scales much better to large numbers of particles. This is consistent with the prior work GATr [2, see Fig. 5 there] from which L-GATr is derived, which shares performance characteristics with L-GATr. Using a message-passing network with sparse connections, which can make it possibly faster than a transformer, is not possible in these particle-physics experiments, because the interaction strength between particles does not decay with the Minkowski distance between their momenta.

In addition to this scaling with the number of particles (or tokens), we also demonstrate in the paper that L-GATr scales well to millions of parameters and to large training datasets (see Fig. 3). Overall, we consider the scalability one of the strongest properties of the model.

&nbsp;

We would like to thank the reviewers again. We hope that we were able to address their questions adequately and look forward to the discussion period.

&nbsp;

**References**:

- [1] A. Butter et al, "Jet Diffusion versus JetGPT - Modern Networks for the LHC", arXiv:2305.10475
- [2] J. Brehmer et al, "Geometric Algebra Transformer", NeurIPS 2023

---

### Decision · Program_Chairs · 2024-09-25

**Decision:**

Accept (poster)

**Comment:**

This paper designs the first Geometric Algebra Transformer (GATr) equivariant to the action of Lorentz group, specifically suited for data of observed particles represented by four-momenta. Reviewers have found this paper well written and well motivated, presents a novel  contribution in its construction of the Lorentz-equivariant transformer architecture, demonstrates (this first example of Lorentz equivariant) generative model for particle physics, and demonstrates several real-life physics applications. On the down side, some reviewers felt the paper not sufficiently accessible to the NeurIPS community, and there were some concerns regarding the computation overhead of the Lorentz transformer layers. The authors have addressed this last point in their rebuttal. Overall, this paper seems like a solid contribution to geometric ML.